# Strength and Deformation Behavior of Graphene Aerogel of Different Morphologies

**DOI:** 10.3390/ma16237388

**Published:** 2023-11-27

**Authors:** Julia A. Baimova, Stepan A. Shcherbinin

**Affiliations:** 1Institute for Metals Superplasticity Problems, Russian Academy of Sciences, Ufa 450001, Russia; 2Higher School of Theoretical Mechanics and Mathematical Physics, Peter the Great St. Petersburg Polytechnic University, Polytechnicheskaya 29, St. Petersburg 195251, Russia; stefanshcherbinin@gmail.com; 3Laboratory “Discrete Models in Mechanics”, Institute for Problems in Mechanical Engineering, Russian Academy of Sciences, St. Petersburg 199178, Russia

**Keywords:** graphene, graphene aerogel, molecular dynamics, mechanical properties

## Abstract

Graphene aerogels are of high interest nowadays since they have ultralow density, rich porosity, high deformability, and good adsorption. In the present work, three different morphologies of graphene aerogels with a honeycomb-like structure are considered. The strength and deformation behavior of these graphene honeycomb structures are studied by molecular dynamics simulation. The effect of structural morphology on the stability of graphene aerogel is discussed. It is shown that structural changes significantly depend on the structural morphology and the loading direction. The deformation of the re-entrant honeycomb is similar to the deformation of a conventional honeycomb due to the opening of the honeycomb cells. At the first deformation stage, no stress increase is observed due to the structural transformation. Further, stress concentration on the junctions of the honeycomb structure and over the walls occurs. The addition of carbon nanotubes and graphene flakes into the cells of graphene aerogel does not result in a strength increase. The mechanisms of weakening are analyzed in detail. The obtained results further contribute to the understanding of the microscopic deformation mechanisms of graphene aerogels and their design for various applications.

## 1. Introduction

The study of tree-dimensional (3D) porous graphene nanostructures is one of the hot issues in different industrial applications. Such structures as aerogels and crumpled graphene are lightweight and low-cost and have high porosity and high strength [1,2]. Among them are graphene aerogels (GA) or cellular honeycomb structures—a promising new class of materials, with properties that can be tuned by designing their morphology. Aerogels are highly porous solid foams that have an interconnected network of thin, solid walls. The details of the atomic arrangement and structural stability of GA were previously intensively studied [3,4,5,6]. Due to their outstanding properties, GA can be used for the fabrication of the composites [7], for electronic devises [8], and for energy harvesting [1,7,9]. The properties of such cellular structures are controlled by their cell geometry and intrinsic morphology [10,11].

One of the most promising methods is the fabrication of GA from graphene oxide, which is synthesized and dispersed in distilled water [12,13]. Graphene and carbon nanotubes have been used to synthesize aerogels with low density and high elasticity [14,15]. Recently, new nanoporous graphene aerogel was obtained by the nanoporous Ni (np-Ni)–based chemical vapor deposition method in which dealloyed np-Ni with 3D bicontinuous open na [16]. To date, the experimental investigation of such structures is commonly based on the understanding of compression behavior, which results in a better description of its main characteristic—high compressibility [17,18,19]. The main challenge is morphology control, for example, creating interactions among constituents or understanding the self-assembly mechanism, which can improve the mechanical and physical properties of GA [20,21]. From this point of view, it is very important to use simulation techniques to understand the basic principles of property control, of the increase in mechanical strength by morphology modifications. Molecular dynamics (MD) is especially effective for such studies and allows one to obtain a large number of new data with a full analysis of the atomic structure of such nanomaterials as GA. The MD method is very good at predicting the mechanical response of graphene, by calculating and summing the evolutions of all atoms. GA involved studies using both MD simulations [22,23,24] and experiments [25,26].

The mechanical properties of graphene aerogel are of high importance for stretchable electronics, wearable devices, and smart manufacturing. Some of the disadvantages of graphene aerogel are the volume shrinkage and the deformation of porous skeletons, which affect the mechanical properties [27,28,29]. GA also show brittle behavior under tension [26]. The poor mechanical strength because of the weak joints in their porous network is another negative characteristic [30]. At the same time, these structures are superelastic and highly compressible [31]. In [32], GA were studied under tension and nanoindentation. The hardness (50.9 GPa) and Young’s modulus (461 ± 9 GPa) were defined from MD simulation. To enhance the mechanical properties of graphene aerogel and make it more stable to deformation, some supporting elements need to be added to the porous of graphene aerogels [33,34,35]. Graphene flakes, which appeared during the synthesis of graphene aerogel or metal nanoparticles, can be added as the reinforcing elements.

Despite the fact that numerous works have been conducted to date on the mechanical behavior of GA, there is still a lack of understanding of the effect of morphology on its strength [26,27,28,29,33,34,35,36]. The change in morphology gives rise to improved mechanical behavior and an increased surface area. Moreover, the majority of works are devoted to the study of compressive behavior in simulation as well as in experiments [37,38,39]. The reason for this is that compressible materials efficiently translate mechanical deformations into electrical signals that have various potential applications. It is desirable to search for high compressive strength and strength recovery, and superb compressibility, in combination with other properties. The systematic study of such factors as morphology, the presence of the reinforcing elements, temperature, and loading direction on the mechanical properties and deformation behavior of graphene aerogels is of high importance.

In the present work, the mechanical properties of three different morphologies of GA under tension are studied by molecular dynamics simulation. The high stretchability is studied and analyzed for tension along different lattice directions. The effect of high temperatures on the strain and deformation behavior is studied.

## 2. Simulation Details

Figure 1 shows three initial cellular structures (part of the simulation cell) under consideration. A part of the simulation cell in an enlarged scale is also presented in Figure 1b as the projection to the xz-plane and in perspective. All of the initial structures are generated by a homemade program, which is used to generate the graphene cellular structure with a different cell morphology. For comparison, graphene flakes and CNTs are added into each honeycomb cell because they are the most studied reinforcement elements for different aerogels [17,40,41,42,43]. Unless otherwise noted, the coordinate system for all structures is the same as the coordinate system shown in Figure 1. In other words, the tension parallel to bridges is equivalent to the tension along the *z* direction, while the tension along lamella is equivalent to the tension along the *x* direction.

The morphology, which will be addressed as a honeycomb is widespread and has been previously fabricated and studied in numerous works [17,26,39,44,45]. This honeycomb-like structure is a geometry that has proven to be highly beneficial to maximizing a bulk-specific elastic modulus [10,46]. Here, the size of the honeycomb wall is *l* = 16 Å, and the number of atoms is 2500.

The morphology, which will be addressed as a re-entrant honeycomb, has been studied in [47,48,49] as a structure with auxetic properties (negative Poisson’s ratio). Here, the main parameters are *a* = 20 Å, *b* = 11 Å (a/b = 0.5), and the inclination angle θ = 60${}^{\circ }$. These parameters were proposed in [47]. The number of atoms in the system is 2448.

One of the possible morphologies of GA, which will be addressed as an arrow honeycomb, is also studied for comparison. Here, the main parameters are *c* = 20 Å, *d* = 11 Å. The number of atoms in the system is 1836. This structure is very close to that discussed in [38,50]. The geometry of the considered structure consists of graphene bridges and lamellae. The bridges are positioned between the lamellae. In the present work, curved graphene nanoribbons (along *z*-axis) can also be called bridges, while long graphene nanoribbons (along *x*-axes) can be called lamellae. The difference with the structure considered in [38] is the curvature of the bridges.

All calculations are performed using the LAMMPS [51,52,53] software package with the AIREBO interatomic potential [54], which has been successfully used to study various properties of a large number of carbon nanosystems, including graphene aerogels [24,55,56,57,58,59], and is proven to reliably reproduce experimentally-obtained mechanical results for graphene [60]. This potential can also be modified to AIREBO-Morse interatomic potential for the description of C-C interactions during compression [61]. Despite the fact that the method of MD is very popular for such studies, especially for the study of the mechanical and fracture behavior of graphene aerogels [55], there are some limitations. For example, under compression, the Lennard–Jones potential contribution to the Adaptive Intermolecular Reactive Empirical Bond Order (AIREBO) potential plays a critical role in predicting early densification. With the increase in the density of GA, the densification of the structure occurs faster with the dissipation energy rising [55]. Despite the fact that AIREBO can effectively describe the realistic mechanical behavior of graphene, it still faces some challenges while reproducing the C-C bond in hybrid structures. In [20,62], Tersoff potential was used as a more suitable option for the simulation of GA.

Despite the slight uncertainty of the AIREBO application for the study of mechanical properties of GA, there is plenty of work devoted to this issue, where simulation is conducted with AIREBO. Based on such works as [37,38,39,63,64], where the deformation behavior of GA was studied and considered in comparison with the experiment, we choose AIREBO interatomic potential for the present work. To avoid non-physical post-hardening behavior under large strains, we set the cut-off distances in the AIREBO potential to 1.92 Å, as suggested by previous studies [37,65,66]. To note, in [67] the deformation behavior of honeycomb graphene was studied with the modified parameters for AIREBO.

In the present work, the tensile behavior of GA is analyzed at two temperatures, 300 and 1000 K, and thus the limitation of the AIREBO potential for the simulation of such effects needs to be discussed. In [68], ReaxFF [69] and AIREBO have been utilized to understand the deformation and fracture mechanics of graphene at the atomic scale to compare the results from DFTB. The other important factor that affects the obtained results is the simulation technique for tension. The deformation under static loading usually shows much higher failure strains than the dynamic loading due to the imposed symmetry [68]. It was also shown that thermal fluctuation under dynamic loading disturbs symmetry and can predict similar failing strains under dynamic loading. Despite having a lot of limitations, the stress–strain curves and failure strain from AIREBO relatively work well.

Periodic boundary conditions are applied in all directions. A Nose–Hoover thermostat is used with a target temperature, a constant number of atoms, and standard velocity–Verlet time integration with a timestep of 1 fs. OVITO is used to visualize the atomistic simulation data. At first, the energy minimization at 0 K is conducted to reach an equilibrium state of the system.

To study the deformation behavior, uniaxial tensile stress is applied along the *x* or *z* axes, and the corresponding stress components are calculated as it is implemented in the LAMMPS package. Stress and energy per atom are also calculated using LAMMPS. At low temperatures, it has been previously shown that the strain rate has only a slight effect on the fracture strength and fracture strain for graphene [70]. Thus, the strain rate of 0.1 fs ${}^{-1}$ was chosen. The length of the simulation cell for all structures is about $L_{x}\;=\;L_{z}$ = 60 Å, $L_{y}$ = 12 Å.

## 3. Results and Discussion

The structural changes during the relaxation were analyzed: the structural parameters were changed insignificantly. For example, sharp edges were smoothed for re-entrant and arrow honeycomb. For arrow honeycomb, cells transform to a more square shape.

The compressive behaviors of such honeycombs under in-plane uniaxial compression was previously studied [37,38]. It was shown that three deformation regimes can be distinguished: (1) stable and nearly uniform deformation and a resulting relatively high stiffness, (2) the coexistence of collapsed and uncollapsed deformation and a resulting essentially zero stiffness, and (3) densification with relatively uniform and stable collapsed deformation and a much stiffer response. Some of the deformation regimes for tension and compression are similar. It is revealed that GA exhibited elastic instability and inelastic collapse during compression.

### 3.1. Effect of Temperature

The effect of temperature on the fracture behavior of GA is studied for all three structures. The tensile behavior was analyzed at 300 K and 1000 K. The thermal stability has been confirmed by MD simulations at temperatures of 300 K and 1000 K [4]. Figure 2, Figure 3 and Figure 4a show stress–strain curves under tension along *x*-axes for honeycomb (Figure 2); re-entrant honeycomb Figure 3; and arrow honeycomb (Figure 4). Figure 2, Figure 3 and Figure 4b present the snapshots of GA as the projection on the xz-plane at critical points. The deformation behavior of GA considerably depends on their morphologies. Additional analysis of the stress per atom (Figure 2, Figure 3 and Figure 4c) and potential energy per atom (Figure 2, Figure 3 and Figure 4d) is conducted for each structure during tension.

Let us consider the deformation process at 300 K in detail for honeycomb GA (see Figure 2). The critical points on the stress–strain curve are labeled as 1–5. No stress changes up to ε = 0.25 occur, which is explained by the simple changes in the shape of honeycomb cells. As seen in Figure 2, initial tension causes the mild flexing of the structure, followed by a series of collapses of cells, leading to an eventual flattening. This can be seen from the snapshots of the structure at critical points (shown by green circles on the black curve). The walls of the honeycomb cells are almost not stressed. The distribution of stress per atom can also be seen from Figure 2c: walls, which are parallel to the tensile direction that became stressed only at ε = 0.5. From the distribution of the potential energy per atom (Figure 2d), the pre-critical strain ε = 0.5 can also be estimated. Up to this value, the potential energy of the system was not increased. The contact of cell walls, which is usually accompanied by sliding and irreversibility, increases the complexities of the constitutive behavior of the honeycomb structure.

Considerable structural changes occur at ε = 0.3, when honeycomb cells are flattened. This collapse mechanism is similar to the CNT collapse. It is well-known that CNT bundles can possess two main stable structural states: open CNTs and collapsed CNTs [71]. Thus, up to ε = 0.25, the structure is not stressed. Further, deformation is mainly defined by the stretching of the walls of honeycomb cells parallel to the tension direction. The initial length of the wall was 16 Åand was unchanged up to ε = 0.25. It was increased to 20 Åbefore the fracture occurred. The analysis of the interatomic bonds showed that on the GNR along the *x*-axis, all bonds *a*, *b*, *c* remained almost unchanged until ε = 0.25. Bonds *a* and *b* changed almost simultaneously (the difference took place due to the thermal fluctuations). These bonds are almost aligned with the loading direction and thus mostly strained. These bonds (and others of the same orientation) are continuously changing from 1.41 Å to 1.75 Å, while bond *c* changes from 1.41 Å to 1.55 Å. Very similar results were obtained previously [72].

As can be seen from Figure 2a, temperature also affects strength and failure strain for honeycomb GA. Previously, it was shown that tensile and compressive behavior considerably depend on the deformation temperature [63]; however, when the temperature is lower than 900 K, the temperature effect is slight, and the in-plane compressive stress–strain curve is similar to the response at room temperature. At higher temperatures, drastic change appears in the in-plane compressive stress–strain curve. The temperature effect can be explained by the larger chemical bond thermal fluctuations and bond length variation at higher temperatures.

Note, in [68] mechanical behavior was analyzed with AIREBO and DFTB for temperature range from 100 to 1000 K. It is found that the critical strain can reduce more than 20% in AIREBO at room temperature; DFTB shows less than a 10% decrease at room temperature. Also, the strength only drops less than 5% with DFTB, but AIREBO shows a 10% drop at room temperature. Such temperature sensitivity of AIREBO comes from the simple bond length criteria. Thus, the obtained results should be analyzed with this correction on the used potential.

For re-entrant honeycomb, just part of the stress–strain curves are presented with critical points labeled as 1–7. As can be seen from the structure snapshots during tension, re-entrant cells begin to open even at the first deformation steps. At ε = 0.1, the shape of the cells is different from the initial and further transforms to a square shape. Again, as for honeycomb, initial tension results in mild flexing of the walls of the structure. A very similar structure with square cells was obtained in [50] and further studied by simulation with a focus on the mechanical stabilizing mechanisms and properties of the deformed structure [38]. At ε = 0.5, honeycomb cells are formed, and the following deformation mechanisms are the same as for the simple honeycomb GA. Similar structural changes have been observed experimentally [44]. Up to ε = 0.5, no changes in the stress per atom and potential energy per atom occur; thus, only structural transformation is presented in Figure 3. Again, temperature slightly affects the stress–strain curves.

As was shown in [38], for such a structure with square cells (called biomimetic), the length of the cell walls (bridges especially) plays a crucial role both in compression and tension: the longer the bridges, the lower the ultimate strength and strain. In the present work, the length of the bridges is equal to the lowest length of bridges from [38]. It was found that GA with short bridges have more bridge-lamellae bonds, so they may absorb more strain than specimens with longer bridges.

As can be seen from Figure 4, temperature slightly affects the tensile strength and failure strain of arrow honeycomb. The critical points on the stress–strain curve were labeled as 1–5. Before relaxation (see Figure 1), there was a sharp angle on the cells oriented along *z*, which was smoothed during relaxation. During tension, the walls of the cells can oscillate, and arrows transform into squares. Cell walls aligned along *x* are mostly stressed since they are oriented along the loading direction.

At the initial state, junctions between graphene nanoribbons are strained a little with the highest concentrated energy (shown by red atoms in Figure 4d). Until ε = 0.04, deformation is determined by the elongation of cell walls oriented along *z*-axis. Until ε = 0.035, stress distribution per atom shows that the cell walls oriented along the *z*-axis are unstrained (Figure 4c), while cell walls oriented along the *x*-axis have higher energies. At the same time, no changes can be seen from energy distribution per atom (Figure 4d). At ε = 0.05 (point 2), all cell walls oriented along the *x*-axis were stressed. At ε = 0.05, the change of the slope on the stress–strain curve is found.

The other change in the slope of the stress–strain curve can be seen at ε = 0.15 (point 3), when all cell walls oriented along the *x*-axis are highly stressed and have much higher potential energy than walls oriented along the *z*-axis. Between ε = 0.15 and ε = 0.2 (point 4), this energy increases even higher, while no changes of stress per atom can be seen. At ε = 0.3 (point 5), there is a square net of GNRs, strained along the *x*-axis. Further, it can be seen that walls aligned along the loading direction have the highest stress and potential energy. The fracture occurs at ε = 3.6, near the junction between walls.

An analysis of the interatomic bonds showed that on the cell walls along the *x*-axis, all of the bonds *a*, *b*, *c* almost remained unhanged until ε = 0.05. Bonds *a* and *b* changed almost simultaneously (the difference took place due to the thermal fluctuations). These bonds are almost aligned with the loading direction and thus mostly strained. These bonds (and others of the same orientation) are continuously changing until ε = 0.2 with further abrupt change until ε = 0.3. At the same time, bond *c* almost does not change until ε = 0.3. Bonds *d*, *e*, *f*, which are analogous to *a*, *b*, *c*, are named differently to distinguish bonds on the cell walls that are differently oriented. These bonds are analyzed on the walls in the loading direction. They slightly change between 1.38 and 1.42 Å due to the thermal fluctuations. Both walls of the cell after relaxation are equal to about 19 Å; however, during tension the cell wall along the *z*-axis did not change, while the length of the cell wall along the *x*-axis increased to 26 Å (increased to 36%).

The deformation mechanisms at 300 K and 1000 K are the same. Strength is decreased just due to the thermal fluctuations.

### 3.2. Effect of Loading Direction

We also have studied the effect of the direction of the applied strain on the deformation behavior of the structure. The uniaxial strain was applied along the *x*- and *z*-axis. Figure 5 presents the stress–strain curves under tension along the *x*- and *z*-axes as the function of applied strain for (a) honeycomb; (b) re-entrant honeycomb; and (c) arrow honeycomb.

Figure 5a shows that for honeycomb GA and re-entrant honeycomb, a significant difference in deformation behavior during tension along armchair (*x*-axis) and zigzag (*z*-axis) was found. It was also previously shown for the same GA that when stretched along the armchair direction, mechanical strength and failure strain decrease with the increase in sidewall width [72]. Moreover, as the sidewall width increases, the strength and ductility gradually decreases. Here, the armchair coincides with the *x*-axis, while the zigzag coincides with the *z*-axis. Note that for re-entrant honeycomb, only part of the stress–strain curve for tension along the *x*-direction is presented, which is explained above. However, the continuation of the stress–strain curve for the re-entrant honeycomb is shown by a dashed line for reference.

For tension along *z*, both structures deformed similarly: the straightening of the walls of honeycomb cells took place with almost no stress concentration; then, stresses over the walls aligned with the tension direction began to increase, which was followed by fracture (analogous to the fracture of arrow honeycomb). For the honeycomb, the length of the wall is about 32 Å, and for the re-entrant honeycomb the length is 22 Å; thus, strength and failure strain for honeycomb GA are higher. Cells for both structures are presented in two structural states for comparison. The same deformation mode with the elongation of the honeycomb cell to the rectangular cell was shown in [63,67]. The obtained values of the fracture strain and strength are also in good agreement with the literature. Thus, the honeycomb GA can be effectively used for the verification of the model accuracy. Interestingly, it was shown in [64] that the resonant frequency of the zigzag honeycomb is lower than that of the armchair honeycomb, which means that the nonlinearity of the zigzag honeycomb is stronger than that of the armchair honeycomb.

For arrow GA, the stress–strain curves for uniaxial tension along the *x* and *z* directions almost coincide. During relaxation, the cells of the arrow honeycomb change their shape into almost square cells. The size of the cell wall along *x* and *z* is 20 Å. Thus, the structure is isotropic in the xz plane. The lower failure strain and tensile strength for the arrow honeycomb stretched along *x* is explained solely by thermal fluctuations. The opened cells of arrow GA are very similar to the opened cells of re-entrant GA.

The addition of CNTs and graphene flakes in the present form does not result in an increase in strength or failure strain. Deformation occurs very differently due to the presence of CNTs and graphene: with these elements, honeycomb cells can transform into square, hexagonal, or collapsed shapes. However, additional studies are required to understand how to control morphology to obtain better mechanical properties of honeycomb. However, reinforcing elements increase the structure density, which results in a strength decrease. Table 1 presents the values of density ρ, tensile strength σ, and failure stress ε for all the considered structures.

As was shown in [16], the strength of the nanoporous graphene intrinsically depends on the density; however, this scaling law depends on the special morphology of the structure. In [16], it was shown that the higher the density, the higher the fracture strength, and this is very close to the behavior of crumpled graphene or paper [37,38]. However, this scaling law can be better used for nanoporous hollow graphene than for honeycomb graphene aerogel. For honeycomb and re-entrant honeycomb, the reverse situation is found: the higher the density, the lower the fracture strain and strength. This is because the key factor affecting the mechanical properties of GA is their microstructure. For GA materials, structural characteristics can vary greatly among the different works. Moreover, the density and mechanical properties of GA prepared by the same method can also vary considerably, while some GA have similar properties despite the use of different production methods [73]. In contrast to [16], the GA materials with lower density and higher mechanical properties were obtained in [74].

### 3.3. Elasticity

These morphologies of GA composed of cells not only possess high strength but also show good elasticity during tension. The elasticity of such structures is of significant interest for the materials used in applications where they are expected to deform periodically while still retaining most of their original strength and stiffness. Let us discuss the elastic behaviors of the considered GA in detail.

During tension in the *x* direction of the arrow honeycomb, the structure remains elastic until the first fracture starts. From the beginning of loading until fracture, there was no change in the atomic configuration, confirming the elasticity of GA. Before ε = 0.15, deformation can be explained by the straightening of the walls of cells. Following the point of initial fracture, the structure changes dramatically and the deformation becomes plastic. During tension in the *z* direction, deformation behavior is very close. Until the sharp increase in stress at ε = 0.25, elastic deformation took place owing to the lack of broken bonds or new close-range non-bonded interactions (see Figure 5). This increase in stress corresponds to pre-critical deformation when all covalent bonds inside the bridges are critically stressed. Interestingly, once the lamellae have buckled globally, new bonds can appear between the neighboring elements, which result in structure stability. Very similar behavior was observed for the other structural morphologies.

## 4. Conclusions

A detailed study has been performed to investigate the mechanical properties of graphene aerogels of a different morphology using MD simulations. The tensile loading of graphene aerogels is analyzed for two temperatures, 300 and 1000 K. Loading is conducted along two lattice directions to overview the structure anisotropy.

At the first deformation stages, the walls (bridges) of the honeycomb cells can easily bend or oscillate in the direction normal to the tension direction. With the strain increase, the stress concentration near the junctions of the honeycomb cell and over the walls oriented along tensile direction can be observed. The strength of GA is considerably dependent on the length of the walls of honeycomb cells and their density. The deformation of the re-entrant honeycomb is defined by the opening of the cells during the first deformation stages. Further, the re-entrant honeycomb transforms into the conventional honeycomb. Very similar deformation mechanisms were found for honeycomb under compression [57]. The new arrow morphology is the weakest since the walls of the cells almost cannot bend. Even at the beginning of the deformation process, the walls (lamellas) are straightened, and the arrows transform into square cells. For all cases, the temperature slightly decreases the strength and failure strain of the GA.

For comparison, aerogel carbon matrices were reinforced by graphene flakes and carbon nanotubes; however, this distribution of the reinforcing elements did not lead to an increase in the strength and failure strain. The search for the new morphologies with special distribution of the reinforcing elements is further required. The obtained results further contribute to the understanding of the microscopic deformation mechanisms of GA and their design for various applications.

## Figures and Tables

**Figure 1 materials-16-07388-f001:**
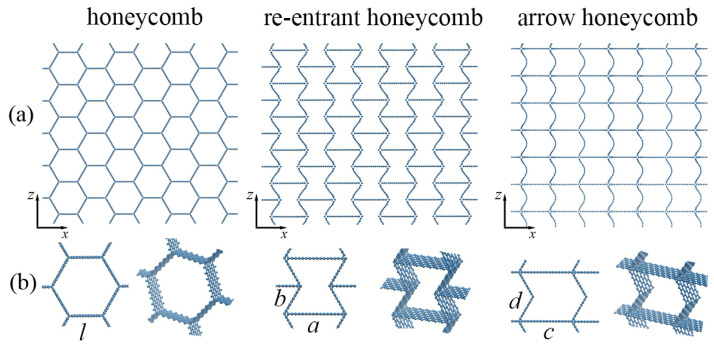
Three morphologies of honeycomb graphene aerogel: honeycomb, re-entrant honeycomb, and arrow-honeycomb. (**a**) Part of the simulation cell in projection to xz plane. (**b**) Part of the simulation cell on an enlarged scale as the projection to xz-plane and in perspective.

**Figure 2 materials-16-07388-f002:**
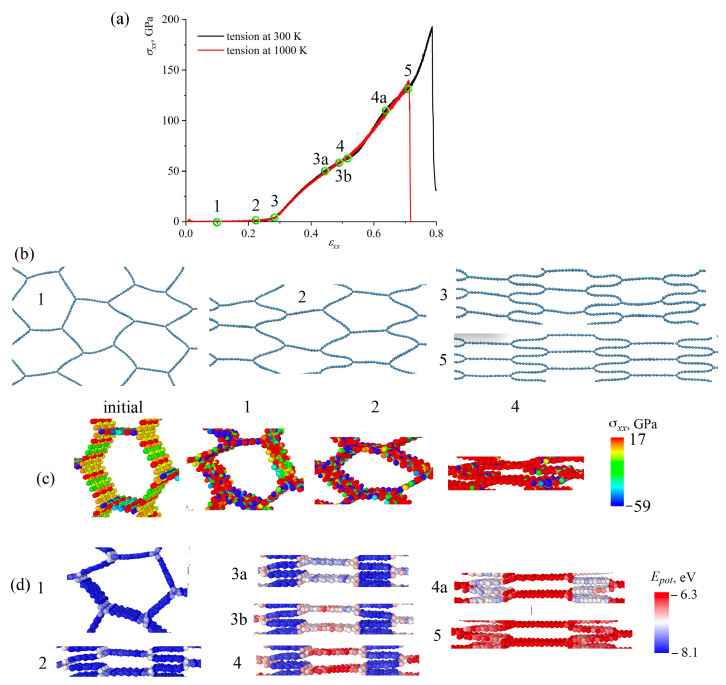
(**a**) Stress–strain curves as the function of strain during tension along *x*-axis for honeycomb GA. The critical points on the stress–strain curve are labeled as 1–5. (**b**) The snapshots of the structure as the projection on xz-plane at critical points for tension at 300 K. Part of the simulation cell is presented. (**c**,**d**) Stress $\sigma_{xx}$ per atom (**c**) and potential energy per atom (**d**) during tension. Part of the simulation cell (one honeycomb cell) is presented.

**Figure 3 materials-16-07388-f003:**
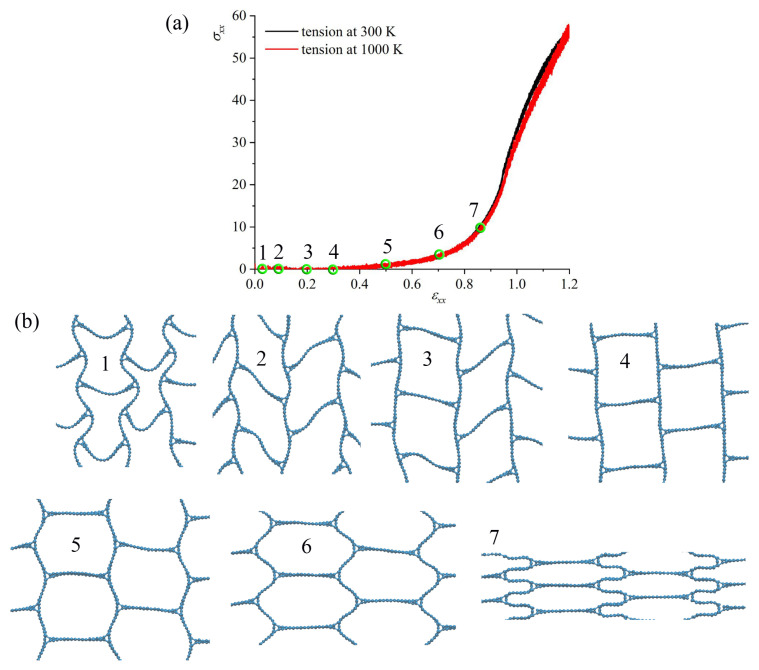
(**a**) Stress–strain curves as the function of strain during tension along *x*-axis for re-entrant honeycomb GA. The critical points on the stress–strain curve are labeled as 1–7. (**b**) The snapshots of the structure as the projection on xz-plane at critical points. Part of the simulation cell is presented.

**Figure 4 materials-16-07388-f004:**
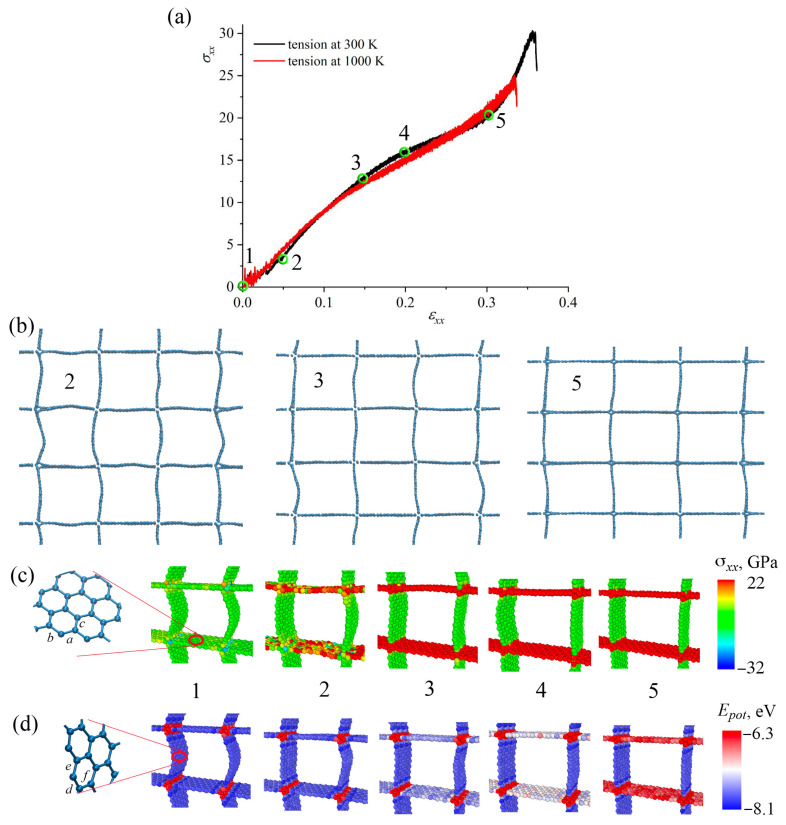
(**a**) Stress–strain curves during tension along *x*-axis for arrow-honeycomb. The critical points on the stress–strain curve were labeled as 1–5. (**b**) The snapshots of the structure as the projection on xz-plane at critical points. Part of the simulation cell is presented. (**c**,**d**) Stress $\sigma_{xx}$ per atom (**c**) and potential energy per atom (**d**) during tension. Part of the simulation cell (one structural element) is presented.

**Figure 5 materials-16-07388-f005:**
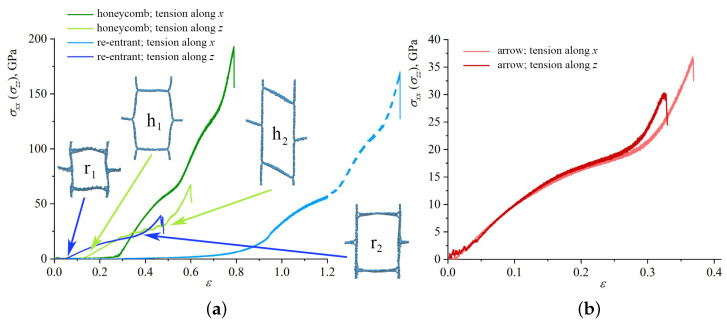
Stress–strain curves as the function of strain during tension along *x*- and *z*-axis for (**a**) honeycomb and re-entrant honeycomb; (**b**) for arrow honeycomb.

**Table 1 materials-16-07388-t001:** Density, tensile strength σ, and failure strain ε for all the considered GA.

		Honeycomb			Re-Entrant			Arrow	
	**GA Matrix**	**Flakes**	**CNT**	**GA Matrix**	**Flakes**	**CNT**	**GA Matrix**	**Flakes**	**CNT**
ρ, g/cm${}^{3}$	0.58	0.86	0.92	1.01	1.31	1.58	0.69	1.21	1.0
σ, GPa	190	156	110	180	160	101	31	27	33
ε	0.79	0.76	0.72	0.8	0.76	0.7	0.36	0.37	0.38

## Data Availability

Data available on request due to restrictions eg privacy or ethical.

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
