# Peer review of "Strength and Deformation Behavior of Graphene Aerogel of Different Morphologies"

_materials, 2023, doi:10.3390/ma16237388_

Round 1

Reviewer 1 Report

Comments and Suggestions for Authors

The manuscript “Strength and deformation behavior of graphene aerogel of different morphology” reports a theoretical study on the aerogel of graphene through molecular dynamics. The study needs to check previous similar studies. There are many modeling studies. Also, they need to carefully check the parameters they used: their capability and limitations. This should be considered in their analysis.

-. Several references need to be clearly cited. e.g., AIREBO [?]

-. Check gyroid/TPMS/nanoporous/3D graphene studies, those are also atomistic models for graphene aerogels. Check this study H. Kashani et al., Science Advances, 2019. They obtained scaling laws from experiments. Please obtain the scaling laws of the authors' models and compare other models in previous studies.

-. Elasticity is also an important property. They can be included with relatively simple calculations.

-. Also, the authors need to modify the cutoff functions of AIREBO. It seems they did not check it. There are many reports discussing the cutoff functions and their stiffening effects.

-. AIREBO has limitation for the temperature effect. It is still fine to see the trend, but it needs to be mentioned that AIREBO overestimates the temperature effects. Check the article “Jung, Gang Seob, Stephan Irle, and Bobby G. Sumpter. "Dynamic aspects of graphene deformation and fracture from approximate density functional theory." Carbon 190 (2022): 183-193.”

Comments on the Quality of English Language

.

Author Response

Thank you very much for kindly editing our manuscript and giving us the opportunity to revise it. We are very grateful for your efforts and valuable suggestions. We have substantially revised the manuscript. The detailed response to the comments is provided below. All changes to the manuscript are highlighted in red. We hope that our revised manuscript and responses will be satisfactory. Thank you very much for your time and consideration.

Comment 1

The study needs to check previous similar studies. There are many modeling studies. Also, they need to carefully check the parameters they used: their capability and limitations. This should be considered in their analysis.

Reply: We appreciate the comment. The Introduction is revised accordingly. We added to the Introduction and Simulation Details Sections some more works we missed previously on similar models and discussed them in the Results Section.

Comment 2

-. Several references need to be clearly cited. e.g., AIREBO [?]

Reply: We thank the reviewer for such a careful reading. References were additionally checked.

Comment 3

Check gyroid/TPMS/nanoporous/3D graphene studies, those are also atomistic models for graphene aerogels. Check this study H. Kashani et al., Science Advances, 2019. They obtained scaling laws from experiments. Please obtain the scaling laws of the authors' models and compare other models in previous studies.

Reply: We thank the reviewer for his advice. The mentioned work was analyzed and discussed in the manuscript. We also analyzed the scaling laws described in that work. A description of the link between the density of the structure and its strength is added to the text.

Comment 4

Elasticity is also an important property. They can be included with relatively simple calculations.

Reply: We thank the reviewer for his comment. We totally agree that the elasticity of the structure is of significant importance. We have added the new subsection, where the elasticity is discussed.

Comment 5

Also, the authors need to modify the cutoff functions of AIREBO. It seems they did not check it. There are many reports discussing the cutoff functions and their stiffening effects.

Reply: We thank the reviewer for this comment. An additional discussion of the cutoff functions of AIREBO is added to the text. Also, the limitations of AIREBO are discussed in the Simulation Details and Introduction sections. The comparison of the obtained results with the literature showed that it is in good agreement with data obtained with different potential functions used. However, it is noted that different simulation techniques were used by different authors for the simulation of tension.

Comment 6

-. AIREBO has limitation for the temperature effect. It is still fine to see the trend, but it needs to be mentioned that AIREBO overestimates the temperature effects. Check the article “Jung, Gang Seob, Stephan Irle, and Bobby G. Sumpter. "Dynamic aspects of graphene deformation and fracture from approximate density functional theory." Carbon 190 (2022): 183-193.”

Reply: We thank the reviewer for his advice. The mentioned work was analyzed and discussed in the manuscript. The limitations of AIREBO are discussed in the Simulation Details and the Results Sections when describing the temperature effect. We also have found other work on the study of temperature effect and compared it with our results.

Reviewer 2 Report

Comments and Suggestions for Authors

In this work, the authors investigate the mechanical properties of three different honeycomb graphene structures, honeycomb, re-entrant honeycomb and arrow honeycomb, without and with reinforcement.

After thoroughly reviewing this manuscript, I don’t recommend publishing it at this stage. I have the following comments and/or suggestions:

1) The manuscript contains several minor grammatical errors.

2) The first part of the manuscript, in which the authors study the mechanical properties of these three structures, is interesting, but I don’t think the second part is relevant. In this second part, the authors investigate the mechanical response of these structures with reinforcement, and the main conclusion is that the reinforcing elements don’t lead to an increase of the strength and the failure strain. In my opinion, this results is not relevant and, thus it isno’t worth publishing in a journal.

Comments on the Quality of English Language

The manuscript contains several minor grammatical errors

Author Response

Thank you very much for kindly editing our manuscript and giving us the opportunity to revise it. We are very grateful for your efforts and valuable suggestions. We have substantially revised the manuscript. A detailed response to the comments is provided below. All changes to the manuscript are highlighted in red. We hope that our revised manuscript and responses will be satisfactory. Thank you very much for your time and consideration.

Comment 1

The manuscript contains several minor grammatical errors.

Reply: We thank the reviewer for such a careful reading. English was additionally checked and grammatical errors were corrected.

Comment 2

The first part of the manuscript, in which the authors study the mechanical properties of these three structures, is interesting, but I don’t think the second part is relevant. In this second part, the authors investigate the mechanical response of these structures with reinforcement, and the main conclusion is that the reinforcing elements don’t lead to an increase of the strength and the failure strain. In my opinion, this results is not relevant and, thus it isno’t worth publishing in a journal.

Reply: We appreciate the comment. The reason for the second part of the study is that the study of reinforcement of different structures by graphene and carbon nanotubes is of high importance now. We search for a way to increase the strength of carbon structures since they can be used as coatings with very different mechanical properties. However, we showed that mechanical strength will depend considerably on the special morphology of graphene cells. Graphene or carbon nanotubes cannot always reinforce carbon structure. Additional discussion is added to prove the need for this second part.

Moreover, we plan to further analyze the effect of especially reinforcing elements. Although here we have no considerable increase of strength with the addition of CNTs or graphene flakes, we are sure, that change of the size/shape/place for reinforcing elements will result in different behavior. The other very good reinforcement is the nanoparticles (metal or silicone), zirconia, and boron nitride, to name a few. This requires considerable attention, and the present work will be continued based on these first results.

Reviewer 3 Report

Comments and Suggestions for Authors

1.       In Introduction, the applications for GA should be in the place before the description for synthesize method. Because the synthesize method is related to the mechanical properties.

2.       References are needed for line 38, 39 and 40.

3.       Why didn’t the author choose experimental method for the study? Is there any limitation for the experimental study?

4.       There may be some format compatibility issue in line 75 and 76.

5.       Why was no failure observed in stress-strain curves in Figure 3a and 6b?

6.       Why didn’t the authors study the compression case? If other people have already done that, I will suggest the authors compare the results to validate the study method.

Comments on the Quality of English Language

Major revision suggested.

Author Response

Thank you very much for kindly editing our manuscript and giving us the opportunity to revise it. We are very grateful for your efforts and valuable suggestions. We have substantially revised the manuscript. A detailed response to the comments is provided below. All changes to the manuscript are highlighted in red. We hope that our revised manuscript and responses will be satisfactory. Thank you very much for your time and consideration.

Comment 1

In Introduction, the applications for GA should be in the place before the description for synthesize method. Because the synthesize method is related to the mechanical properties.

Reply: We agree with the reviewer. The introduction part was corrected.

Comment 2

References are needed for line 38, 39 and 40.

Reply: We agree with the reviewer. References were added. 

Comment 3

Why didn’t the author choose experimental method for the study? Is there any limitation for the experimental study?

Reply: We appreciate the comment. To date, there are almost no limitations in the experimental fabrication of honeycomb graphene aerogels, however, the experiment is always more expensive than simulation work. Our simulation allows us to obtain some recommendations for how to generate such a honeycomb structure and achieve better properties. Simulation allows us to analysis three different GA morphologies – honeycomb, re-entrant honeycomb (which is already known), and arrow honeycomb which is new. Two different reinforcement elements were added, thus the number of studied morphologies is increased. Moreover, mechanical behaviour was analyzed on the basement of atomic movement which is very important for understanding the mechanical properties of carbon nanomaterials. Additional explanations are added to the Introduction.

Comment 4

There may be some format compatibility issue in line 75 and 76.

Reply: We thank the reviewer for such a careful reading. Fixed.

Comment 5

Why was no failure observed in stress-strain curves in Figure 3a and 6b?

Reply: We appreciate the comment. In Figure 3a and 6b stress-strain curves for re-entrant honeycomb are presented. For the re-entrant honeycomb, just part of the stress-strain curves is presented since only at this stage the mechanical behavior is different from that of the honeycomb. As can be seen from the structure snapshots during tension, re-entrant cells begin to open even at the first deformation steps. At epsilon=0.1, the shape of the cells is different from the initial and further transforms to a square shape. At epsilon=0.5, honeycomb cells are formed, and the following deformation mechanisms are the same as for the honeycomb GA. However, we present the full stress-strain curve for the re-entrant honeycomb in Fig. 5 (shown by a blue dashed line) for reference. As it can be seen it is totally the same as for honeycomb and there is no need to analyze it again.

Comment 6

Why didn’t the authors study the compression case? If other people have already done that, I will suggest the authors compare the results to validate the study method.

Reply: We appreciate the comment. Compression behavior was previously studied and thus we did not simulate this. However we agree with the reviewer and additional discussion and comparison with the literature were added to the text in the Introduction, Simulation Details and Results Sections.

Round 2

Reviewer 2 Report

Comments and Suggestions for Authors

I still believe that the second part of the paper is not publishable. Otherwise, the authors have revised the manuscript according to my questions

Author Response

Thank you very much for your kindly reviewing our manuscript. We are very grateful for your efforts and valuable suggestions. We have substantially revised the manuscript. 

Comment: I still believe that the second part of the paper is not publishable. Otherwise, the authors have revised the manuscript according to my question

Reply:

In accordance with your valuable suggestion, we decided to remove the second part; however, the first reviewer asked to analyze the effect of structure density on its mechanical properties. Thus, we cannot totally remove these results, but we removed the figure and wrote just one short paragraph on the effect of morphology and density. We removed subsection 3.4. We also shorten the simulation details accordingly. 

Reviewer 3 Report

Comments and Suggestions for Authors

I suggest accepting in the present form.

Author Response

Thank you very much for your kindly reviewing our manuscript. We are very grateful for your efforts and valuable suggestions. Thank you very much for your time and consideration.